# The Effect of a Care Bundle on the Rate of Blood Culture Contamination in a General Intensive Care Unit

**DOI:** 10.3390/antibiotics13111082

**Published:** 2024-11-13

**Authors:** Fani Veini, Michael Samarkos, Pantazis-Michael Voutsinas, Anastasia Kotanidou

**Affiliations:** 1Infection Prevention Unit, Evaggelismos Hospital, 10676 Athens, Greece; enl@evaggelismos-hosp.gr; 21st Department of Internal Medicine, Medical School, National & Kapodistrian University of Athens, Laikon Hospital, 11527 Athens, Greece; 3Department of Pathophysiology, Medical School, National & Kapodistrian University of Athens, Laikon Hospital, 11527 Athens, Greece; pantazis88@gmail.com; 41st Intensive Care Department, Medical School, National & Kapodistrian University of Athens, Evaggelismos Hospital, 10676 Athens, Greece; akotanid@med.uoa.gr

**Keywords:** blood culture, blood specimen collection, equipment contamination, intensive care unit, quality improvement, patient care bundles

## Abstract

**Background/objectives**: Blood culture (BC) contamination is a frequent problem which leads to increased laboratory workload, inappropriate use of antibiotics and the associated adverse events, and increased healthcare costs. This study prospectively examined the effect of a care bundle on BC contamination rates in a high workload ICU. **Results:** During the study, in total, 4236 BC vials were collected. After the intervention, the BC contamination rate decreased significantly from 6.2% to 1.3%. The incidence rate of contaminated BC sets was significantly lower following the intervention: 0.461 vs. 0.154 BC sets per 100 ICU bed-days. Overall compliance with the BC care bundle increased dramatically from 3.4% to 96.9%. **Methods:** We performed a before–after study in a general ICU from January 2018 to May 2019, with the intervention starting on November 2018. Blood culture sets were classified as positive, contaminated, indeterminate, and negative. We used bivariate and interrupted time series analysis to assess the effect of the intervention on BC contamination rates and other BC quality indicators. **Conclusions:** The BC care bundle was effective in reducing BC contamination rates and improving several quality indicators in our setting. The indeterminate BC rate is an important but understudied problem, and we suggest that it should be included in BC quality indicators as well. A significant limitation of the study was that the long-term effect of the intervention was not assessed.

## 1. Introduction

According to the 2022–2023 European Centre for Disease Prevention and Control (ECDC) Point Prevalent Study, bloodstream infections (BSI) represent almost 18.0% of all healthcare-associated infections (HAIs) in intensive care units (ICUs) across Europe [1]. It is therefore important that BSIs be diagnosed accurately.

Although novel, non-culture, methods for the detection of bacteremia are evolving, blood cultures (BCs) remain the gold standard against which all new methods are compared [2,3]. Timely and accurate reporting of microbiologic data from positive BCs improves clinical outcomes and reduces healthcare costs [4]. In fact, obtaining BCs before antibiotic administration is one of the five elements of the “hour-1 bundle for initial resuscitation for sepsis and septic shock” [5]. Thus, BCs are among the most frequently performed and clinically important tests in microbiology. Unfortunately, contamination of BCs is a frequent problem, with reported rates ranging from 0.8% to 30%, depending on procedure-specific factors (e.g., appropriate antisepsis, appropriate blood volume) and patient-specific factors (age, race, body mass index) [6,7,8,9]. Blood culture contamination causes several important problems, such as additional unnecessary testing (i.e., to confirm the presence of the commensal organism and to clarify whether a BSI is actually contaminated or not); increased antimicrobial exposure, leading to the emergence of antimicrobial resistance and adverse effects such as allergic reactions and *C. difficile* infection; and, finally, inappropriate hospital admissions, leading to increased laboratory workload and healthcare costs [6,10]. Several interventions to reduce BC contamination are supported by evidence, e.g., appropriate selection and preparation of the venipuncture site, sterile techniques during venipuncture and inoculation of the blood in the BC vials, collection of an appropriate blood volume, and expedited transport to the laboratory [6,9,11]. Most of these interventions can be introduced in clinical practice using a care bundle. A bundle is a structured way of improving the processes of care and patient outcomes: a small, straightforward set of evidence-based practices—generally three to five—that, when performed collectively and reliably, improve the delivery of healthcare and patient outcomes [12,13].

Although there are relatively few studies on BC collection care bundles, they are suggested as a quality improvement intervention [14,15,16,17]. The success of care bundles is, at least in part, setting-dependent, and requirements in terms of the organizational culture are probably critical to successful care bundle implementation. Thus, a multidisciplinary approach, with collaboration among clinicians, laboratory personnel, and infection control teams can enhance bundles’ effectiveness [13]. Intensive care units (ICUs) present a unique environment with higher rates of BC contamination compared with general wards due to the high workload of staff, the high severity and urgency of illness, and the poor vascular condition of patients [18,19]. Patients in critical condition, or with sepsis or acute kidney injury, are at increased risk of BC contamination [7], and hospitalization in the intensive care unit (ICU) is an independent risk factor for BC contamination [8]. The impact of BC contamination in ICU patients may be larger, since unnecessary antimicrobial exposure leads to adverse effects, *C. difficile* infection, and the emergence of antimicrobial resistance, which have probably more consequences in these vulnerable patients.

Taking into account that Greece ranks above the 75th percentile of the EU/EE countries, with 69.4 BC sets drawn per 1000 patient-days [20], we designed this study to prospectively examine the effect of the implementation of a care bundle on BC contamination rates in our setting, an ICU with a 91.2% occupancy rate and 9.731 patient-days per year, with high rates of BCs and limited experience in the implementation of care bundles.

## 2. Results

### 2.1. Patient Characteristics

Data were collected from 419 patients in the pre-intervention phase (PRE, 1 January 2018 to 31 July 2018) and from 328 patients in the post-intervention phase (POST, 1 November 2018 to 31 May 2019). The demographic and clinical characteristics of the patients are presented in Table 1. Blood cultures (BCs) were drawn from 255/419 (60.9%) patients in PRE, and from 211/328 (64.3%) patients in POST (chi squared = 0.944, *p* = 0.33). Mortality rates were similar in both phases (37.8% in PRE and 34.1% in POST), as were the Charlson, APACHE II, and SOFA scores upon admission. The only difference between the study phases was the frequency of wound drainage tubes, which was larger in the PRE phase (PRE, 102/419 (24.3%) vs. POST, 59 (18.0%), *p* = 0.036). The rates of infections between the two phases were similar.

### 2.2. Blood Culture Information

During the study, in total, 4236 BC vials were collected (2050 in the PRE and 2186 in the POST phase). Of the 1763 complete BC sets, 989/1763 (56.1%) were paired BC sets (464 in the PRE and 525 in the POST phase), while 443/4236 (10.4%) vials were solitary. Details of the BC yield are shown in Table 2. The most common pathogen isolated was coagulase-negative staphylococci (124/511, 24.3% of all isolates), followed by *K. pneumoniae* (110/511, 21.5%) and *A. baumanii* (80/511, 15.7%). Details are provided in Appendix A. The most common indication for obtaining BCs overall was increased CRP (1647/2206 BC sets, 74.7%), followed by either leukocytosis, leukopenia, or neutropenia (1432/2206, 64.9%). Fever was the indication for a BC in only 904/2206 (41.0%) BC sets, while in 80/2206 (3.6%) BC sets, there was no clear indication. Some indications were significantly more frequent in the PRE phase (central venous catheter change and a previously positive BC), while increased CRP was more frequent in the POST phase (see Appendix A for details).

### 2.3. Contaminated and Indeterminate Blood Cultures

There were 314/1763 (17.8%) positive BC sets and 61/443 (13.8%) positive solitary BC vials. Of the 114 BC sets with a common commensal (5.2% of all BC sets), 36 (1.6%) were classified as contaminated BC sets (CBCs) and 78 (3.5%) as indeterminate BC sets (IBCs) (see Table 2). The contamination rate decreased significantly after the intervention from 6.2% (29/464) to 1.3% (7/525) (chi-square = 16.98, *p* < 0.0001, relative risk = 0.21 (95% confidence interval: 0.09–0.47)). The proportion of indeterminate BC sets was also significantly lower in the POST phase (PRE, 65/1210 (5.2%) vs. 15/996 (1.5%), relative risk = 0.28 (95% confidence interval: 0.16–0.50), chi-square = 21.93, *p* < 0.001). The contamination rate by pooling CBC and IBC sets was again lower for the POST phase (PRE, 92/1210 (7.6%) vs. 22/996 (2.2%), relative risk = 0.28 (95% CI: 0.17–0.50), chi-square = 32.44, *p* < 0.0001).

The duration of the study was relatively short for a robust interrupted time series analysis (ITS). However, a preliminary ITS analysis, which included CBC and IBC sets, showed that there was a significant change in the trend after the implementation of the intervention (from 1.38 to 0.47, difference = −0.91, 95% CI: −1.46 to −0.36, Figure 1). The supremum Wald test confirmed the presence of a change point in the series (*p* = 6.03 × 10^−9^).

The incidence rate of CBC sets was significantly lower in the POST phase: 0.461 vs. 0.154 BC sets per 100 ICU bed-days (rate difference = −0.307, 95% confidence interval = −0.527 to −0.086). The indeterminate BC incidence rate was also significantly lower in the POST phase:1.00 vs. 0.330 (rate difference = −0.671, 95% confidence interval = −0.996 to −0.347).

Overall, CBC and IBC sets represented, respectively, 7.4% (36/489) and 16.0% (78/489) of BC sets yielding any microorganism (pathogen or commensal). The proportion of CBC sets was significantly lower in the POST phase: 29/308 (9.4%) vs. 7/181 (3.9%), chi-square 5.14, *p* = 0.023. Similarly, the proportion of IBC sets was also lower in the POST phase: 63/308 (20.4%) vs. 15/181 (8.3%), chi-square 12.59, *p* = 0.0004.

### 2.4. Quality Indicators

The proportion of paired BC sets increased significantly after the intervention from 464/882 (52.6%) to 525/881 (59.6%) (chi-square = 8.7, *p* = 0.0031, relative risk = 1.13 with 95% confidence interval = 1.04–1.23). The proportion of solitary BC vials decreased significantly from 328/2050 (16.0%) to 115/2186 (5.3%), chi-square = 130.0, *p* < 0.0001.

The appropriate blood volume per BC vial is at least 8 mL [22,23]; however, the proportion of BC vials with appropriate blood volume was only 709/2050 (34.6%) in the PRE phase. This proportion increased significantly to 2024/2186 (93.5%) in the POST phase (chi-square = 1614.0, *p* < 0.001). The median vial blood volume also increased significantly from 6.40 mL (SD = 2.61) to 8.73 mL (SD = 1.04), *t*-test, *p* < 0.0001. Further details on the blood volume per vial are shown in Appendix A.

### 2.5. Care Bundle Compliance

Compliance with the care bundle was assessed by direct observation of randomly selected BC collection events in both the PRE and POST phases. Overall, we collected 275 observations: 145 in the PRE (100 for venipuncture and 45 for central venous catheter (CVC) blood sampling) and 130 in the POST phase (95 for venipuncture and 35 for CVC). The overall compliance in the PRE phase was very low at 3.4% (5/145), but it increased to 96.9% (126/130) in the POST phase (Fisher’s exact test, *p* < 0.0001). Details of compliance by study phase and bundle element are in Appendix A.

### 2.6. Factors Associated with BC Contamination

To analyze risk factors for BC contamination, we used a case–control design, in which the negative BC sets were the controls. We ran two analyses, one in which we compared CBC sets with negative BC sets and a second one in which we compared the combined CBC and IBC sets with negative BC sets. The first analysis was necessary, as CBCs are definitely contaminated BC sets. The second analysis was carried out to study the effect of the IBCs, which are usually not included in analyses of contamination of BCs, although, in several cases, IBCs might be truly contaminated BCs. Positive sets (BC sets *n* = 375) were not included in any risk factor analyses [7].

Blood culture set-specific risk factors

For the analysis of different indications for BCs, we included all BC sets for each patient (*n* = 1831). There was no association between CBCs and the different indications for BCs. However, the risk of CBCs or IBCs was lower with increased CRP (74/1363 (5.4%) vs. 40/468 (8.5%) relative risk = 1.40, 95% CI: 1.07–1.80, chi-square = 5.8 *p* = 0.02). In contrast, the risk of CBCs or IBCs was higher in previous positive BCs (34/402 (8.5%) vs. 80/1428 (5.6%), relative risk = 1.11, 95% CI: 1.006–1.28, chi-square = 4.4 *p* = 0.036). The complete results are shown in Appendix A. Contaminated BC sets were associated with lower BC volumes (CBCs, median = 6.0 mL (interquartile range = 3.0 mL) vs. negative BCs, median = 8.0 mL (interquartile range = 4.0 mL), Mann–Whitney test *p* = 0.007). There was no association between contamination and the interval in days between hospital or ICU admission and the BC day (see Appendix A for details).

Patient-specific risk factors

Patient-specific variables included demographic variables, underlying diseases and comorbidities, pharmaceutical and other interventions, clinical status assessment scales (e.g., APACHE, SOFA), and laboratory parameters (e.g., albumin and C-reactive protein). For these variables, we planned to analyze only the initial BC set for each patient [8]. This dataset included 423 patients (236 from the PRE phase and 187 from the POST phase), of whom 6/423 (1.4%) had contaminated initial BCs, 16/423 (3.8%) had indeterminate initial BC results, and 401/423 (94.8%) had a negative initial BC set. However, the analysis of CBCs vs. negative BC sets did not reveal any associations, possibly due to the low percentage of CBCs (details in Appendix A). We repeated the analysis for CBCs or IBCs vs. negative BCs. We found that only patients on chronic hemodialysis had a significantly higher risk of CBCs or IBCs: 4/13 (23.5%) vs. 18/388 (4.4%), Fisher’s exact test *p* = 0.008, odds ratio = 6.6 (95% CI 1.96–22.38). No other significant association was found (see Table 3 and Figure 2).

## 3. Discussion

In the present study, we found that the introduction of the bundle to the study ICU was associated with a significant reduction in the BC contamination rate from 6.2% to 1.3%. Our results are very similar to those of Kai et al., who implemented a BC care bundle in the emergency department [14]. Minami et al., in a retrospective study, found a similar effect after the implementation of a BC care bundle across a large hospital [15]. Our study was conducted exclusively in an ICU; thus, one may conclude that BC care bundles are effective across practically all hospital departments. Although the study had the limitations of the before–after design, the effect of the care bundle was confirmed by significant reductions in the incidence rate (per 100 ICU bed-days) of both CBC and IBC sets, and by a preliminary ITS analysis (Figure 1).

The intervention was associated with improvement in all other BC quality indicators, such as the proportion of paired and complete BC sets, the proportion of solitary BC vials, the proportion of BC vials with an appropriate blood volume, and the mean vial blood volume (see the Section 2). The overall compliance with the BC care bundle increased enormously after the intervention from 3.4% to 96.9%. This increase is too large to be real. Since the observation of compliance with the BC bundle was overt, it is possible that the behavior of the healthcare workers under observation changed, i.e., there was a Hawthorne effect [24]. A recent study in an ICU found that the difference between overt and covert observations of hand hygiene compliance could be as high as 25% [25]. It should also be noted the POST period was relatively short (November 2018 to May 2019) and it did not allow us to assess how sustained the impact of the intervention was. Unfortunately, we could not reassess compliance with the BC care bundle after the end of the study.

It is important to note that the study ICU was characterized by a baseline BC contamination rate of 6.2%, which is clearly above the acceptable range of <3% [2,6]. The low overall compliance with the BC bundle, the many solitary vials (328/2050, 16.0%), and the low proportion of paired BS sets (56.1%) in the PRE phase all point to poor BC collection practices. Another characteristic of our setting was the increased proportion of IBC sets (5.2%, 63/1210) at baseline, which may represent a problem that is not fully recognized. The management of a CBC set is rather straightforward, as most laboratories do not perform susceptibility testing, and the clinician can relatively safely conclude that there is no infection, at least from isolated common commensal [26,27]. In contrast, the management of an indeterminate BC result requires a decision based on clinical judgement, and this might lead to a full BC workup with susceptibility testing and subsequent antimicrobial use [7]. To our knowledge, there is no literature on the significance of IBC sets. In our study, the number of IBC sets was much larger than the CBC sets, but we had no detailed data regarding antimicrobial use, and thus we could not investigate possible differences between patients with CBC and IBC sets. Indeterminate classification is a consequence of the BC set being single; therefore, the proportion of single BC sets might be another important indicator of BCs’ collection quality. Current CDC guidance suggests the calculation of the single-set BC rate at least monthly in conjunction with the CBC rate, as a quality sub-measure [28]. A similar suggestion has been made specifically for the emergency department by Hills et al. [29]. In our opinion, the indeterminate BC set rate, calculated as the number of indeterminate BC sets divided by the number of single-set BCs, might be more useful than the single-set BC rate, as it expresses more accurately the BC sets that might lead to excess laboratory workload and antimicrobial use.

Our study was not designed to study risk factors for BC contamination, and the small sample size and the low number of outcomes suggest that the power of the study was inadequate to establish associations with patient-specific factors [7,8]. Furthermore, we had no data regarding BC collection practices and compliance with the BC care bundle for individual BC collection events. Therefore, we could not assess these variables as risk factors for contamination. The only BC-specific risk factors that we could assess were the indication for BC collection and the blood volume. We found that lower blood volumes were associated with CBC sets, as reported in the literature [30].

Our study has several limitations; perhaps the more important is that the duration of the study did not allow us to have enough data time points to perform a robust ITS analysis. However, a preliminary ITS analysis suggested that the improvement in the outcomes studied was indeed associated with the implementation of the BC care bundle. Second, we had no data to associate BC bundle compliance in an individual BC collection procedure with the presence of contamination in the collected BC sets. This has been documented by Minami et al. in a retrospective study [15]. Third, the ICU studied had poor BC collection quality indicators at baseline; therefore, the margin for improvement was very large. The observed effects of the intervention might not be expected across settings with, e.g., better baseline BC collection quality. Finally, we were not able to document the duration and the size of the effect of the intervention in the long term.

## 4. Materials and Methods

### 4.1. Setting

The study was conducted at the intensive care unit (ICU) of Evaggelismos Hospital, a 945-bed university-affiliated tertiary care hospital in Athens, Greece. The ICU has 30 beds, with an 89% bed occupancy rate and 9.750 patient-days per year.

### 4.2. Study Design

The study was quasi-experimental with a before–after design: a pre-intervention phase (PRE, 1 January 2018 to 31 July 2018), an implementation interval (1 August 2018 to 31 October 2018), and a post-intervention phase (POST, 1 November 2018 to 31 May 2019). To analyze the risk factors for BC contamination, we used a case–control design, where CBCs or CBC plus IBC sets were classified as cases, and the negative BC sets were the controls.

The study protocol was approved by the Institutional Review Board of the hospital (No. 302/04-12-2017).

### 4.3. Inclusion/Exclusion Criteria

All BCs obtained from patients > 18 years old in the ICU during the PRE and POST phases were included in the study. If multiple BCs had been collected from one patient, they were all included. Blood cultures obtained from patients before they were transferred to the ICU were excluded.

### 4.4. Data Collection

Clinical data were collected from the medical records and included demographic data, comorbidities, the presence of various devices (such as vascular stents, etc.), therapeutic interventions (e.g., immunosuppressive drugs), vital signs, and laboratory results at different time points. We also calculated the Acute Physiology and Chronic Health Evaluation II (APACHE II) and the Sequential Organ Failure Assessment (SOFA) scores on admission and on the day of blood culture.

### 4.5. Intervention: Care Bundle Implementation

To support the implementation of the care bundle, we performed various educational activities for the healthcare workers (physicians, nurses, and nursing assistants) of the ICU. The activities were in-person training sessions, focused on the correct technique of obtaining BCs and the “care bundle to obtain BCs”. These activities were supported by written material, which included a detailed protocol for obtaining BCs, a leaflet outlining the care bundle and giving detailed instructions on how to draw a BC, and the care bundle checklist (see Appendix A). We adopted the “care bundle to obtain blood cultures” of Antimicrobial Resistance and Healthcare Associated Infection Scotland, with some modifications for BCs from CVCs [17]. The elements of the care bundle are presented in Table 4.

### 4.6. Definitions and Outcomes

We defined as a set all BC vials drawn during the same venipuncture, including at least one aerobic vial. “Complete” was any BC set which included at least two vials. When only anaerobic or fungal BC vials were collected, they were classified as “solitary” BC vials. When a second BC set was obtained within 48 h of the initial BC set, the set was defined as “paired”; if not, the BC set was defined as “single”.

The BC sets which yielded a common commensal were considered contaminated following the CDC guidance; however, we extended the time frame for the repeat BC to 48 (instead of 24) hours [28]. Pathogenic microorganisms and common commensals were classified on the basis of the National Healthcare Safety Network Organisms List [31].

A set was classified as positive, negative, contaminated, or indeterminate. A “positive” BC set was defined as any BC set yielding a pathogenic microorganism, or all BC sets drawn within 48 h from the same patient, yielding the same commensal microorganism. A BC set was classified as “negative” when no vial yielded any microorganism. A “contaminated” BC set (CBC) was defined as any set with at least one BC vial (aerobic, anaerobic, or fungal) of a set yielding a common commensal, provided that (a) the BC set was paired, and (b) the particular organism had not been isolated from another of these BC sets. An “indeterminate” BC set (IBC) was defined as any BC set yielding a common commensal when the set was single [32].

The primary outcome was the contamination rate, calculated as the number of CBC sets divided by the number of paired BC sets [28]. Secondary outcomes included the IBC set rate and the pooled CBC or IBC set rate (both in relation to all BC sets obtained), the incidence rate of CBC and IBC sets per 100 ICU bed-days, the proportion of CBC and IBC sets as a percentage of BC sets which yielded a microorganism, and compliance with the care bundle (overall and for each element). We compared all outcomes in the PRE and POST phases.

To investigate the risk factors for contamination among BC set-specific variables (e.g., indication, day of ICU hospitalization, blood volume) we analyzed BC sets, excluding positive and indeterminate BC sets. For patient-specific variables (e.g., age, sex, underlying diseases, etc.) we analyzed only the initial BC set of each patient (REF). Again, we excluded positive BC sets; however, as the number of patients with contaminated initial BC sets was low (*n* = 6), we performed the analysis both excluding indeterminate initial BC sets and pooling contaminated and indeterminate initial BC sets.

### 4.7. Statistical Analysis

For descriptive statistics, for continuous variables, we calculated the means and standard deviations (SD) or medians and interquartile ranges (IQR), as appropriate. For categorical variables, we calculated counts and percentages. We used Pearson’s chi-square test or Fisher’s exact test for nominal variables and 2-sample independent *t*-tests or the non-parametric Mann–Whitney test for continuous variables to investigate associations between the outcome and the different variables, as appropriate. The incidence rates of events per patient-days were calculated and compared using Poisson’s rates. Interrupted time series analysis was performed using the “Robust Interrupted Time Series” (RITS) toolbox [21]. Significance levels were set at a two-tailed *p*-value of 0.05. Statistical analysis was performed using IBM SPSS Statistics version 26.0 and StatsDirect version 4.04.

## 5. Conclusions

The implementation of a BC care bundle was associated with a significant improvement in the BC contamination indicators, as well as with improvements in almost all other BC collection quality indicators. The expected benefits of the improvement in the BC contamination rate in an ICU might be higher than that in a general ward. As ICU patients have higher baseline rates of BC contamination and of antibiotic exposure, the expected reductions in antibiotic exposure and possibly cost would be larger in absolute terms.

It is noted that the poor baseline BC collection practices in the study ICU suggest that this effect should not be expected across all settings. In addition, the long-term benefits of the intervention were not studied. An important finding was the large number of indeterminate BC sets, which is associated with the respective proportion of single-set BC. Indeterminate BC sets may have an equal or a larger impact than CBC sets, and we suggest that they should be included in the BC collection quality indicators. The study adds to the body of evidence supporting the effectiveness of care bundles, and highlights their importance in infection control in high-risk settings, such as an ICU.

## Figures and Tables

**Figure 1 antibiotics-13-01082-f001:**
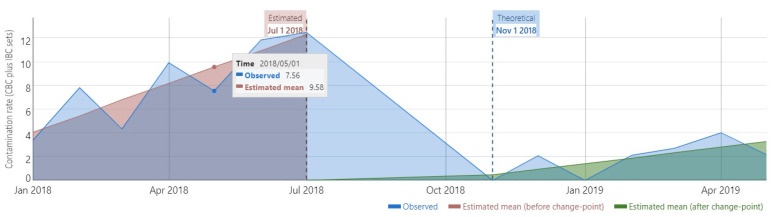
Interrupted time series (ITS) analysis of the contamination rate (including both CBC and IBC sets). The graph represents the monthly contamination rate throughout the study period. The method used for the ITS takes into account the start of the intervention (theoretical change point) but, according to the data, estimates the actual change point (estimated change point), as described by Cruz et al. [21].

**Figure 2 antibiotics-13-01082-f002:**
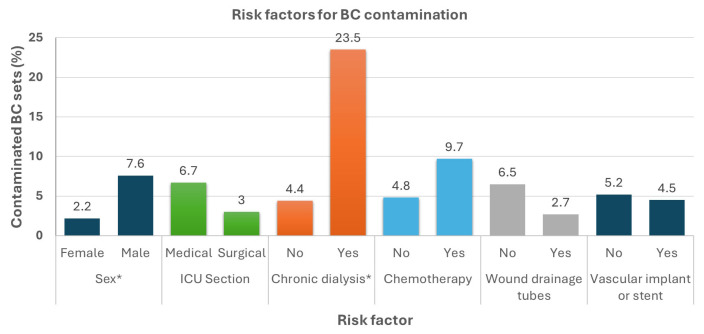
Selected possible risk factors for BC contamination. Each bar represents the proportion of patients with a contaminated BC. Only sex and chronic dialysis were significantly associated with BC contamination. For details, see Table 3. * Chi-square, *p* = 0.008.

**Table 1 antibiotics-13-01082-t001:** Sample characteristics in each phase.

Characteristic	PRE (*n*, (%))	POST (*n*, (%))	*p* Value
Age, (mean, SD)	61.2 (16.7)	60.5 (16.9)	0.711 ^‡^
Male gender	238 (56.8)	210 (64)	0.054 ^†^
Patients with blood culture	255 (60.9)	211 (64.3)	0.331 *
Hospital admission to ICU, days (median, IQR)	3 (0–7)	3 (1–10)	0.091 ^§^
Days in ICU, (median, IQR)	6 (2–19)	7 (2–18)	0.313 ^§^
Days in hospital, (median, IQR)	22 (11–47)	27 (12–56.5)	0.131 ^§^
Death during hospitalization	158 (37.8)	112 (34.1)	0.303 *
Charlson Comorbidity Index,	0 (0–4)	0 (0–4)	0.733 ^§^
Prosthetic heart valve	7 (1.7)	5 (1.5)	0.875 *
Pacemaker	8 (1.9)	5 (1.5)	0.690 *
Vascular implant or stent	15 (3.6)	12 (3.7)	0.954 *
Orthopedic endoprosthesis	5 (1.2)	3 (0.9)	>0.999 ^†^
APACHE II at admission (median, IQR)	10 (0–17)	10 (0–17.5)	0.547 ^§^
SOFA at admission (median, IQR)	6 (0–9)	6 (0–10)	0.668 ^§^
Short-term central venous catheter	242 (57.8)	202 (61.6)	0.290 *
Tunneled central venous catheter	0 (0)	3 (0.9)	0.084 ^†^
Biliary or urinary drainage or stent	4 (1)	8 (2.4)	0.109 *
Wound drainage tubes	102 (24.3)	59 (18)	0.036 *
Peritoneal dialysis catheter	1 (0.2)	0 (0)	>0.999 ^†^
Urinary bladder catheter	250 (59.7)	216 (65.9)	0.083 *
Intubation	223 (53.2)	187 (57)	0.302 *
Continuous renal replacement therapy	68 (16.2)	46 (14)	0.406 *
Respiratory infection	102 (24.3)	62 (18.9)	0.075 *
Urinary tract infection	3 (0.7)	4 (1.2)	0.705 ^†^
Intra-abdominal infection	19 (4.5)	9 (2.7)	0.201 *
Skin and soft tissue infection	18 (4.3)	7 (2.1)	0.103 *
CNS infection	7 (1.7)	4 (1.2)	0.763 ^†^
Surgical site infection	1 (0.2)	1 (0.3)	>0.999 ^†^

* Pearson’s chi-square test; ^†^ Fisher’s exact test; ^‡^ Student’s *t*-test; ^§^ Mann–Whitney test. SD, standard deviation; IQR, interquartile range; APACHE, Acute Physiology and Chronic Health Evaluation; SOFA, Sequential Organ Failure Assessment; CNS, central nervous system. The listed infections represent clinically diagnosed syndromes for which blood cultures were obtained.

**Table 2 antibiotics-13-01082-t002:** Blood culture yield by study phase.

Blood Culture Sets			
	PRE Phase, *n* (%)	POST Phase, *n* (%)	Total, *n* (%)
**Positive**	164 (18.6%)	150 (17%)	314 (17.8%)
Monomicrobial	138 (15.6%)	140 (15.9%)	278 (15.8%)
Polymicrobial	26 (2.9%)	10 (1.1%)	36 (2%)
**Common commensal**	73 (8.3%)	21 (2.4%)	94 (5.3%)
Contaminated	26 (2.9%)	7 (0.8%)	33 (1.9%)
Indeterminate	47 (5.3%)	14 (1.6%)	61 (3.5%)
**Negative**	645 (73.1%)	710 (80.6%)	1355 (76.9%)
**Subtotal**	882 (100%)	881 (100%)	1763 (100%)
Solitary vials			
	PRE phase, *n* (%)	POST phase, *n* (%)	Total, *n* (%)
**Positive**	52 (15.9%)	9 (7.8%)	61 (13.8%)
Monomicrobial	44 (13.4%)	6 (5.2%)	50 (11.3%)
Polymicrobial	8 (2.4%)	3 (2.6%)	11 (2.5%)
**Common commensal**	19 (5.8%)	1 (0.9%)	20 (4.5%)
Contaminated	3 (0.9%)	0 (0%)	3 (0.7%)
Indeterminate	16 (4.9%)	1 (0.9%)	17 (3.8%)
**Negative**	257 (78.4%)	105 (91.3%)	362 (81.7%)
**Subtotal**	328	115	443
All sets			
	PRE phase, *n* (%)	POST phase, *n* (%)	Total, *n* (%)
**Positive**	216 (17.9%)	159 (16%)	375 (17%)
Monomicrobial	182 (15%)	146 (14.7%)	328 (14.9%)
Polymicrobial	34 (2.8%)	13 (1.3%)	47 (2.1%)
**Common commensal**	92 (7.6%)	22 (2.2%)	114 (5.2%)
Contaminated	29 (2.4%)	7 (0.7%)	36 (1.6%)
Indeterminate	63 (5.2%)	15 (1.5%)	78 (3.5%)
**Negative**	902 (74.5%)	815 (81.8%)	1717 (77.8%)
**Total**	1210	996	2206

**Table 3 antibiotics-13-01082-t003:** Patient-related risk factors for CBC or IBC sets.

Risk Factors for Contaminated BCs	Category	Contaminated BC Sets, *n* (%)	Negative BC Sets, *n* (%)	Chi-Square, *p* Value	Odds Ratio (95% CI)
Age (mean, SD)	n/a	59.0 (53.0–69.0)	64.0 (47.0–74)	0.75	n/a
Sex	Female	4 (2.2)	182 (97.8)	0.14	3.74 (1.24–11.24)
	Male	18 (7.6)	219 (92.4)		
ICU section	Medical	17 (6.7)	238 (93.3)	0.118	0.43 (0.15–1.19)
	Surgical	5 (3)	163 (97)		
Charlson Comorbidity index ^†^	n/a	2.0 (1.0–6.0)	3.0 (1.0–5.0)	0.91	n/a
Chronic dialysis	No	18 (4.4)	388 (95.6)	0.008	6.63 (1.97–22.38)
	Yes	4 (23.5)	13 (76.5)		
Asplenia	No	22 (5.2)	398 (94.8)	1.0	n/a
	Yes	0 (0)	3 (100)		
HIV (no AIDS)	No	22 (5.3)	396 (94.7)	1.0	n/a
	Yes	0 (0)	5 (100)		
Bone marrow transplant	No	22 (5.3)	392 (94.7)	1.0	n/a
	Yes	0 (0)	9 (100)		
Solid organ transplant	No	22 (5.3)	395 (94.7)	1.0	n/a
	Yes	0 (0)	6 (100)		
BMI < 18.5	No	22 (5.3)	397 (94.7)	1.0	n/a
	Yes	0 (0)	4 (100)		
Prosthetic heart valve	No	22 (5.3)	392 (94.7)	1.0	n/a
	Yes	0 (0)	9 (100)		
Pacemaker	No	22 (5.3)	392 (94.7)	1.0	n/a
	Yes	0 (0)	9 (100)		
Vascular implant or stent	No	21 (5.2)	380 (94.8)	1.0	0.86 (0.11–6.72)
	Yes	1 (4.5)	21 (95.5)		
Orthopedic endoprosthesis	No	22 (5.3)	393 (94.7)	1.0	n/a
	Yes	0 (0)	8 (100)		
APACHE II on admission ^†^		16.0 (11.0–23.0)	15.00 (11.0–20.0)	0.44	
SOFA on admission ^†^		9.0 (8.0–11.0)	8.0 (7.0–11.0)	0.29	
SOFA on BC day *^†^		9.0 (6.0–11.0)	9.0 (5.5–11.0)	0.65	
Albumin on admission ^†^		3.3 (2.6–3.9)	3.0 (2.5–3.6)	0.19	
CRP on admission ^†^		9.2 (1.0–31.5)	8.4 (2.0–19.1)	0.48	
Albumin on BC day *^†^		2.5 (2.2–3.2)	2.6 (2.2–3.1	0.94	
CRP on BC day *^†^		8.1 (6.3–18.7)	13.1 (6.4–18.3)	0.30	
Chemotherapy	No	19 (4.8)	373 (95.2)	0.21	2.1 (0.59–7.54)
	Yes	3 (9.7)	28 (90.3)		
Corticosteroids	No	20 (5.2)	365 (94.8)	1.0	1.01 (0.23–4.51)
	Yes	2 (5.3)	36 (94.7)		
Short-term CVC	No	0 (0)	28 (100)	0.38	n/a
	Yes	22 (5.6)	373 (94.4)		
Implantable CVC	No	22 (5.2)	398 (94.8)	1.0	n/a
	Yes	0 (0)	3 (100)		
Biliary or urinary drainage or stent	No	22 (5.3)	390 (94.7)	1.0	n/a
	Yes	0 (0)	11 (100)		
Wound drainage tubes	No	18 (6.5)	257 (93.5)	0.109	0.39 (0.13–1.19)
	Yes	4 (2.7)	144 (97.3)		
Peritoneal dialysis catheter	No	22 (5.2)	399 (94.8)	1.0	n/a
	Yes	0 (0)	2 (100)		
Urinary bladder catheter	No	0 (0)	7 (100)	1.0	n/a
	Yes	22 (5.3)	394 (94.7)		
Intubation	No	2 (3.4)	56 (96.6)	0.75	1.62 (0.37–7.14)
	Yes	20 (5.5)	345 (94.5)		
CRRT	No	18 (5.6)	302 (94.4)	0.61	0.68 (0.22–2.05)
	Yes	4 (3.9)	99 (96.1)		
Respiratory infection	No	17 (6.1)	260 (93.9)	0.26	0.54 (0.19–1.50)
	Yes	5 (3.4)	141 (96.6)		
Urinary tract infection	No	22 (5.3)	396 (94.7)	1.0	n/a
	Yes	0 (0)	5 (100)		
Intra-abdominal infection	No	22 (5.5)	379 (94.5)	0.61	n/a
	Yes	0 (0)	22 (100)		
Skin and soft tissue infection	No	22 (5.5)	379 (94.5)	0.61	n/a
	Yes	0 (0)	22 (100)		
CNS infection	No	21 (5.1)	392 (94.9)	0.41	2.07 (0.25–17.14)
	Yes	1 (10)	9 (90)		
CR-BSI	No	22 (5.2)	400 (94.8)	1.0	n/a
	Yes	0 (0)	1 (100)		
Surgical site infection	No	22 (5.2)	399 (94.8)	1.0	n/a
	Yes	0 (0)	2 (100)		

^†^ Median (25th–75th percentile), * data for *n* = 148 patients. n/a: not applicable.

**Table 4 antibiotics-13-01082-t004:** Elements of the blood culture bundle.

Bundle for Blood Cultures from Venipuncture	Bundle for Blood Cultures from CVC *
Disinfect the cap of BC vials with 70% alcohol	Disinfect the cap of BC vials with 70% alcohol
Perform hand hygiene before procedure	Perform hand hygiene before procedure
Use of CHG 2% for the skin, leave time to dry	Use gloves (sterile or not, it depends)
Use an aseptic technique, no touching critical sites	Scrub the hub with CHG 2% for 30 s
Inoculate BC vials first (before other tests)	Inoculate BC vial first, follow other tests

* CVC = central venous catheter.

## Data Availability

The data presented in this study are available on request from the corresponding author due to legal restrictions.

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
