# Peer review of "The Effect of a Care Bundle on the Rate of Blood Culture Contamination in a General Intensive Care Unit"

_antibiotics, 2024, doi:10.3390/antibiotics13111082_

Round 1
Reviewer 1 Report
Comments and Suggestions for Authors
In the manuscript, the authors conducted a before-and-after study to evaluate the short-term effects of a care bundle on the rate of blood culture contamination within a general ICU. It is logical to expect reductions in the incidence rate in the short term, and I appreciate that the authors have acknowledged the study's limitations.
I suggest providing additional details regarding the methodology to enhance the interest of readers. Specifically, it would be beneficial to include the written materials, such as the comprehensive protocol for obtaining blood cultures (the care bundle for blood culture collection), as well as the educational resources supplied to healthcare personnel as supplementary documentation. Furthermore, the inclusion of more data concerning isolated pathogenic organisms.
Thank you.
Comments on the Quality of English LanguageThe English could be improved to more clearly express the research.
Author Response
REVIEWER 1
“I suggest providing additional details regarding the methodology to enhance the interest of readers. Specifically, it would be beneficial to include the written materials, such as the comprehensive protocol for obtaining blood cultures (the care bundle for blood culture collection), as well as the educational resources supplied to healthcare personnel as supplementary documentation. Furthermore, the Inclusion of more data concerning isolated pathogenic organisms.”
- Unfortunately, the requested educational material is in Greek and we had no time to translate them in English. We have included the requested materials as non-published material (BC leaflet & instructions, BC checklist, BC Knowledge Questionnaire). We suggest that the Editor decides whether this material is included in the Supplementary material or not.
- Regarding the pathogenic organism isolated, we have included a comment in the text (L102-4 of the revised MS) and a Supplementary table with the most frequent pathogens.
Reviewer 2 Report
Comments and Suggestions for Authors
Veini et al. have conducted a well-designed study. The article is well-written and shows that the care bundle was successful to reduce contamination of blood cultures in the short-term. The authors also have a good knowledge of appropriate literature, and have contextualised their study well throughout. I have comments for minor revisions, mostly regarding clarity.
Specific comments below.
Line 44: What are the other factors that affect blood culture contamination?
Line 44-45: “multiple unwanted effects” – passive language, please revise. I encourage the authors to be more specific here and highlight the necessity of BCs and why it is so important to avoid BC contamination.
Line 46: “spurious” is not the correct term here. Perhaps “idiopathic” is better. Furthermore, I encourage the authors to briefly mention why it is so critical to have accurate results from BCs: use of the incorrect antimicrobial can lead to increased microbial resistance; patient health may decline with incorrect treatment; overuse of antibiotics in a patient can also lead to further issues like C. difficile infection, etc.
Line 60: Can the authors be more specific regarding what is meant by “a high workload ICU”?
Line 61: there is an extra “.”
Line 64: Although the authors have defined “PRE” and “POST” in the methods, the methods are at the end of the article. Therefore, I encourage the authors to briefly describe what is meant here.
Table 1: It may be not be immediately clear to the reader whether the infections listed at the end of the table are co-occurring or infections that have resulted in blood infection. Can the authors briefly clarify this in a footnote?
Line 73: It is not immediately clear what is meant by the footnote “all values are n (%)”. The formatting in Table 3 is best, where the authors have titled the columns with what the numbers are referring to, eg. “Contaminated BC set, n (%)”. I recommend the authors also apply this format to Table 2 in addition to Table 1.
Line 92: Abbreviations must be defined at their first use. “CBC” and “IBC” are not defined until the methods, which are at the end of the article.
Figure 1: Please appropriately label the y axis. Also, it is not clear what is meant by “Estimated mean functions, Unit Evaggelismos Hospital ICU”. Please also provide more detail in the figure legend: for example, sample size, how these values were attained, what analysis was carried out. Also please clearly define “estimated” and “theoretical” in the figure legend. Furthermore, there are unnecessary lines on the graph, please remove these.
Line 110: Please be consistent with how numbers are formatted, eg use a “.” for a decimal point, not “,”
Line 135: “Details of compliance by study phase and bundle element are in supplementary” – the “are” is missing here
Lines 144 to 153: these should be indented to indicate the info belongs to the bullet points, same with the paragraph below “patient-specific risk factors”
Line 160: Please be consistent with use of abbreviations. For example, “CBC” and “IBC” are used mostly, but occasionally “contaminated BC” or “indeterminate BC” is used. Please define CBC and IBC at their first use in the article, and then be consistent throughout. It may be appropriate to re-define these abbreviations in new sections of the article.
Table 3: Could the authors also add a figure to help visualize risk factors for BC contamination?
Lines 187 to 194: It is not immediately clear what the authors mean here. Please clarify what is meant by Hawthorne effect. Is it possible for the authors to do more statistical analysis to take into account/correct for any possible Hawthorne effect? For clarity, in the sentence “It should also be noted the POST period was relatively short”, can the authors add in how long the POST period was?
Line 271: CVCs have not been defined at their first use.
Author Response
REVIEWER 2
Line 44: What are the other factors that affect blood culture contamination?
- Factors added in L45-47.
Line 44-45: “multiple unwanted effects” – passive language, please revise. I encourage the authors to be more specific here and highlight the necessity of BCs and why it is so important to avoid BC contamination.
- Revised as requested – L47-52
Line 46: “spurious” is not the correct term here. Perhaps “idiopathic” is better
- Paragraph revised – see above, L47-52.
Line 60: Can the authors be more specific regarding what is meant by “a high workload ICU”?
- We have included ICU bed occupancy rate and admissions per year – L78
Line 61: there is an extra “.”
- Deleted
Line 64: Although the authors have defined “PRE” and “POST” in the methods, the methods are at the end of the article. Therefore, I encourage the authors to briefly describe what is meant here.
We apologize for that – Definitions of PRE and POST included in section 2.1, L82-83.
Table 1: It may be not be immediately clear to the reader whether the infections listed at the end of the table are co-occurring or infections that have resulted in blood infection. Can the authors briefly clarify this in a footnote?
- These are infections for which blood culture was obtained. Clarified in the footnote of Table 1.
Line 73: It is not immediately clear what is meant by the footnote “all values are n (%)”. The formatting in Table 3 is best, where the authors have titled the columns with what the numbers are referring to, eg. “Contaminated BC set, n (%)”. I recommend the authors also apply this format to Table 2 in addition to Table 1.
- We have changed the format of Table 2 as requested.
Line 92: Abbreviations must be defined at their first use. “CBC” and “IBC” are not defined until the methods, which are at the end of the article.
- Abbreviations defined at section 2.3, L117.
Figure 1: Please appropriately label the y axis. Also, it is not clear what is meant by “Estimated mean functions, Unit Evaggelismos Hospital ICU”. Please also provide more detail in the figure legend: for example, sample size, how these values were attained, what analysis was carried out. Also please clearly define “estimated” and “theoretical” in the figure legend. Furthermore, there are unnecessary lines on the graph, please remove these.
- Figure 1 has been modified as requested. We have included the missing labeling on axis y and deleted the “Estimated mean functions etc.”. We have explained briefly the method in the Figure legend and provided a reference.
Line 110: Please be consistent with how numbers are formatted, eg use a “.” for a decimal point, not “,”
- We apologize for that – This is a constant issue for us using software in Greek and English language. All decimal points were changed to “.”.
Line 135: “Details of compliance by study phase and bundle element are in supplementary” – the “are” is missing here.
- “are” added.
Lines 144 to 153: these should be indented to indicate the info belongs to the bullet points, same with the paragraph below “patient-specific risk factors”
- Lines in paragraphs “Blood culture specific risk factors” and “Patient-specific risk factors” indented.
Line 160: Please be consistent with use of abbreviations. For example, “CBC” and “IBC” are used mostly, but occasionally “contaminated BC” or “indeterminate BC” is used. Please define CBC and IBC at their first use in the article, and then be consistent throughout. It may be appropriate to re-define these abbreviations in new sections of the article.
- We have defined CBC and ICB on their first occurrence in the text and use them throughout. Please note that in L181 we did not use the abbreviation for Contaminated BC, as it is suggested that abbreviations should avoided at the beginning of a sentence )AMA Manual of style).
Table 3: Could the authors also add a figure to help visualize risk factors for BC contamination?
- We have added Figure 2, in which selected risk factors for BC contamination have been included.
Lines 187 to 194: It is not immediately clear what the authors mean here. “It should also be noted the POST period was relatively short”, can the authors add in how long the POST period was?
- We have briefly explained Hawthorne effect (L222-24) and included the dates of the POST period (L227).
Line 271: CVCs have not been defined at their first use.
- Defined in section2.5 (L160).
Reviewer 3 Report
Comments and Suggestions for Authors
This manuscript offers valuable insights into blood culture contamination; however, addressing a few concerns and incorporating suggested improvements will enhance its quality prior to publication. These refinements will ensure clarity, strengthen the flow, and increase the overall impact of the findings.
1. Abstract Revision: Streamline the language for clarity and cohesion, ensuring a logical flow of sentences that concisely summarize the study’s objectives, methodology, and findings. Correct punctuation and eliminate redundancies.
2. Introduction: Explanation of Care Bundles
- Introduce care bundles as structured sets of evidence-based practices designed to improve patient outcomes by standardizing processes that reduce contamination risk.
3. Clarify Line 32
- Specify whether the 18% rate includes contaminated BCs or is a distinct measurement.
4. Abbreviations in Table 1
- Define all abbreviations at the bottom of Table 1 to improve readability.
5. ICU Context for Contamination
- Explain contamination risks in ICUs, emphasizing patients’ vulnerability due to compromised immune systems and the risks from unnecessary antibiotic exposure. Highlight the importance of a multidisciplinary approach, as collaboration among clinicians, lab staff, and infection control teams can enhance bundle effectiveness.
6. Define PRE and POST
- Clearly define "PRE" and "POST" upon first mention and ensure all abbreviations are explained consistently throughout.
7. Clarify Analysis in Section 2.6
- Explain the rationale for analyzing CBC sets separately and for combined CBC and IBC sets, clarifying the distinction between these analyses. Is not well written.
8. Restructure First Paragraph of Results
- Begin with a concise summary of primary findings, then discuss previous studies to provide context, highlighting any unique findings or notable differences.
9. Expanded Conclusion
- Summarize the implications of reduced BC contamination in ICUs, acknowledging study limitations and suggesting future research directions. Emphasize the broader impact of care bundles on infection control in high-risk environments.
10. The manuscript requires careful proofreading to improve sentence flow and correct punctuation, ensuring that ideas connect smoothly and clearly.
Sincerely,
Comments on the Quality of English Language
The manuscript would benefit from a thorough proofreading to improve sentence flow and correct punctuation errors. Ensuring smooth transitions and accurate punctuation will enhance readability and clarity, strengthening the overall quality of the work.
Author Response
REVIEWER 3
- Abstract Revision: Streamline the language for clarity and cohesion, ensuring a logical flow of sentences that concisely summarize the study’s objectives, methodology, and findings. Correct punctuation and eliminate redundancies.
- Abstract revised according to the suggestions
- Introduction: Explanation of Care Bundles- Introduce care bundles as structured sets of evidence-based practices designed to improve patient outcomes by standardizing processes that reduce contamination risk.
- A brief definition of care bundles with an appropriate reference was included (L56-60)
- Clarify Line 32 - Specify whether the 18% rate includes contaminated BCs or is a distinct measurement.
- Unfortunately, there was no such information in the source (ECDC Report).
- Abbreviations in Table 1- Define all abbreviations at the bottom of Table 1 to improve readability.
- We have defined all abbreviations in the footnotes of Table 1.
- ICU Context for Contamination - Explain contamination risks in ICUs, emphasizing patients’ vulnerability due to compromised immune systems and the risks from unnecessary antibiotic exposure. Highlight the importance of a multidisciplinary approach, as collaboration among clinicians, lab staff, and infection control teams can enhance bundle effectiveness.
We have included a brief discussion of these issues in the Introduction, L66-74.
- Define PRE and POST - Clearly define "PRE" and "POST" upon first mention and ensure all abbreviations are explained consistently throughout.
- PRE and POST periods defined in section 2.1, L82-83.
- Clarify Analysis in Section 2.6 - Explain the rationale for analyzing CBC sets separately and for combined CBC and IBC sets, clarifying the distinction between these analyses. Is not well written.
- We have included an explanation for the rationale of the inclusion of IBC in the analysis (L167-72).
- Restructure First Paragraph of Results - Begin with a concise summary of primary findings, then discuss previous studies to provide context, highlighting any unique findings or notable differences.
- We believe the reviewer means the first paragraph of the Conclusions, which have restructured as suggested (L205-214.)
- Expanded Conclusion - Summarize the implications of reduced BC contamination in ICUs, acknowledging study limitations and suggesting future research directions. Emphasize the broader impact of care bundles on infection control in high-risk environments.
We have added text in the Conclusion, mentioning these two issues (L354-7 & L364-5).
- The manuscript requires careful proofreading to improve sentence flow and correct punctuation, ensuring that ideas connect smoothly and clearly.
- We have done our best to improve language and punctuation in the text.